# Prediction of the Current and Future Distribution of Tomato Leafminer in China Using the MaxEnt Model

**DOI:** 10.3390/insects14060531

**Published:** 2023-06-06

**Authors:** Hangxin Yang, Nanziying Jiang, Chao Li, Jun Li

**Affiliations:** 1Key Laboratory of Prevention and Control of Invasive Alien Species in Agriculture & Forestry of the North-Western Desert Oasis, Ministry of Agriculture and Rural Affairs, College of Agronomy, Xinjiang Agricultural University, Urumqi 830052, China; 13565949023@163.com (H.Y.); 15276661895@163.com (N.J.); 2Guangdong Key Laboratory of Animal Conservation and Resource Utilization, Guangdong Public Laboratory of Wild Animal Conservation and Utilization, Institute of Zoology, Guangdong Academy of Sciences, Guangzhou 510260, China; 3Western Agricultural Research Center, Chinese Academy of Agricultural Sciences, Changji 831100, China

**Keywords:** *Tuta absoluta*, potential suitable habitat, climate change

## Abstract

**Simple Summary:**

Tomato leafminer (*Tuta absoluta*), an important quarantine pest in China, was first detected in China in Yili, Xinjiang Uygur Autonomous Region, in 2017. Its damage has grown in recent years, severely harming Solanaceae plants in China and causing enormous economic losses. Here, tomato leafminer’s potential distributions in China under the current climate and four future climate models (SSP1–26, SSP2–45, SSP3–70, and SSP5–85) were predicted using the maximum entropy (MaxEnt) model with ArcGIS software, and the accuracy of the prediction results was tested. Under the four models, the area increased in the suitable region of China, with the smallest increase in SSP1–26 and the greatest increase in SSP5–85. Combined with the prediction results of this study, we can clearly see the future diffusion trend of the tomato leaf moth in China, which has important theoretical significance for the control of the tomato leaf moth in China.

**Abstract:**

Tomato leafminer (*Tuta absoluta*), an important quarantine pest in China, was first detected in China in Yili, Xinjiang Uygur Autonomous Region, in 2017. Its damage has grown in recent years, severely harming Solanaceae plants in China and causing enormous economic losses. The study and prediction of the current and future suitable habitats for tomato leafminer in China can provide an important reference for the monitoring, early warning, and prevention and control of the pest. Here, tomato leafminer’s potential distributions in China under the current climate and four future climate models (SSP1–26, SSP2–45, SSP3–70, and SSP5–85) were predicted using the maximum entropy (MaxEnt) model with ArcGIS software, and the accuracy of the prediction results was tested. The areas under the receiver operating characteristic curves of the models were all greater than 0.8, and the test omission rate of the model simulation results basically agreed with the theoretical omission rate, suggesting that the prediction results had satisfactory accuracy and reliability. Under the current climatic conditions, the highly suitable habitats for tomato leafminer in China are mainly distributed in most of North China, most of East China, most of South China, most of Central China, most of Southwest China, some parts of Northeast China, and only a few parts of Northwest China. Annual mean temperature is the main environmental factor limiting the distribution. The suitable habitats for tomato leafminer will shift under different future climate models: Under SSP1–26, the highly suitable habitats will spread to the north and northeast and to the southeast coastal areas; under SSP2–45, the size of highly suitable habitats will grow from the present to 2080 and shrink from 2081 to 2100; under SSP3–70, the highly suitable habitats will spread northeastwards, but the highly suitable habitats in southeast coastal areas will shrink from 2081 to 2100 and turn into moderately suitable habitats. Under SSP5–85, the highly suitable habitats will spread northeastwards and northwestwards, with the size of highly suitable habitats gradually decreasing and the size of moderately suitable habitats increasing. Different climates will lead to different distributions of suitable habitats for tomato leafminer, with annual mean temperature, isothermality, and mean diurnal range as the main environmental influences.

## 1. Introduction

Tomato leafminer (*Tuta absoluta*) (Meyrick) (*Lepidoptera: Gelechiidae*) is a destructive invasive pest native to South America. Since its invasion of Spain in 2006, it has spread to Europe, the Mediterranean region, and, in 2010, to some countries in Central Asia and Southeast Asia [1]. Tomato leafminer has now infested nearly 90 countries and regions, causing severe damage to local tomato industries [2]. This pest has been found in several countries bordering the northwest and southwest of China and it has shown a trend of further spread. The pest was first detected in Yili Prefecture, Xinjiang Uygur Autonomous Region, China, in August 2017 [3]. It was also found in Midu County, Dali Prefecture, Yunnan, in March 2018 [4]. Tomato leafminer moth can cause 80–100% reduction in tomato yield [5]. This pest poses a great threat to the tomato cultivation and the development of the tomato industry [6].

Tomato leafminer is characterized by rapid reproduction, strong dispersal ability, and highly overlapping of generations [5]. The pest mainly harms tomatoes of the Solanaceae family but does non-negligible harm to crops such as potatoes, eggplants, peppers, tobacco, and nightshade. Tomato leafminer harms crops mainly through its larvae, which feed directly by infiltrating leaves, stems, and fruits but can also damage crops indirectly by inducing fruit rot and leaf blight [7]. Females prefer to lay eggs on new leaves to facilitate the infiltration of the larvae after hatching [8]. Once the fruit is damaged by this pest, the loss of economic value can be as high as 80–100% [9]. As the largest tomato-producing country, China has witnessed rapid development of its tomato processing industry, which has given farmers ways to generate revenue from agricultural products and increase their income. Xinjiang Uygur Autonomous Region is the largest tomato production base in China, accounting for more than 70% of the country’s production and an annual export volume of 700,000 tons, and economic losses could reach 2.9 billion dollars if the pest were to inflict damage without control. Therefore, it is crucial to monitor, prevent, and control tomato leafminer [10].

Studies have shown that temperature is the main factor affecting the flight displacement, activity time, and speed of insects [11] Temperature and humidity are key factors that can determine habitat range and population size [12]. Temperature and humidity not only affect the growth and development of insects, but also interfere with the growth and development of insect host plants, indirectly affecting the diffusion of insect populations. Due to the fact that the growth and development of insects are mainly influenced by environmental factors such as temperature and humidity, the authors’ establishment of the MaxEnt model mainly considers environmental variables that are related to temperature and humidity.

The prediction of suitable habitat distribution is an important part of monitoring invasive alien species. The main models used for predicting and forecasting suitable habitats for invasive alien species include BIOCLIM, CLIMEX, the genetic algorithm for rule-set production (GARP), and maximum entropy (MaxEnt) models [13]. The MaxEnt model does not rely on the species’ biological parameters, needs no species missing points, avoids extensive field work, is little affected by sample size bias, and, thus, can achieve good predictive accuracy even with a limited distribution of data [14,15,16]. Therefore, the MaxEnt model has been widely used in the fields of invasive species prediction, species conservation, and pest control [17,18]. The MaxEnt model is currently the most effective ecological niche model when used in combination with geographical information systems (GIS) and has high predictive accuracy [19,20].

There have been reports on the spread of suitable habitats for tomato leafminer. Tonnang et al., and Xian et al., used the CLIMEX model to predict the potential geographic distribution of the pest in Africa and China, respectively [1,21]. Liu et al., used the MaxEnt model and the CLIMEX model to predict the potential global geographic distribution and overwintering boundary of tomato leafminer [19]. Luo et al., predicted the adaptability of tomato leafminer in China, but they did not optimize the MaxEnt model for prediction and selected only three coupled models to predict the suitable habitats for tomato leafminer up to 2080 [22,23].

In the present study, the distribution points and environmental variables of tomato leafminer were screened based on published data, and the MaxEnt model was optimized by adjusting its feature combination and regularization multiplier parameters. After optimization, the potential suitable habitats for tomato leafminer in China were predicted using four different shared socioeconomic pathways (SSPs) representing different future climate models, so as to provide a reference and theoretical basis for the development of reasonable control measures and to offer a basis for vigilance against the spread and invasion of tomato leafminer in various places.

## 2. Materials and Methods

### 2.1. Collection and Processing of Distribution Data of Tomato Leafminer

The tomato leafminer distribution data used in this study were obtained from three sources: (1) The species distribution data from the valid monitored data record model of tomato leafminer were downloaded from the global biodiversity information facility (GBIF) database (http://www.gbif.org/ accessed on 18 May 2022); (2) Data were collected from the literature [4,24,25,26,27]; (3) The data publicly reported and recorded by Chinese governments were also collected.

After manual deletion of preserved specimen, the distribution point data were screened. To reduce oversampling in some areas and undersampling in others, we reused the ENM Tools software to screen distribution point data, with a screening accuracy of 5 km, and 393 effective distribution points data were obtained (Figure 1). The standard China base map is derived from the Standard Map Service (http://bzdt.ch.mnr.gov.cn/index.html accessed on 22 May 2022) of the Ministry of Natural Resources, with a ratio of 1:4 million.

### 2.2. Selection of Environmental Variables

Shared socioeconomic pathways (SSPs) are part of a new scenario framework established by the climate change research community to facilitate comprehensive analysis of future climate impacts, vulnerability, adaptation, and mitigation [28]. New hypothetical scenarios for future human carbon emissions and changes in socioeconomic factors in the 21st century were proposed in the international Coupled Model Intercomparison Project Phase 6 (CMIP6). The distribution of suitable habitats for tomato leafminer under the influence of future climate was predicted using four climate change scenarios (SSP1–26, SSP2–45, SSP3–70, and SSP5–85) [29].

Data for the environmental variables in the global prediction model were derived from the 19 bioclimatic variables and one environmental factor in WORLDCLIM version 2.1 (http://www.worldclim.org/): annual mean temperature (bio01), mean diurnal range (bio02), isothermality (bio03), temperature seasonality (bio04), maximum temperature of the warmest month (bio05), minimum temperature of the coldest month (bio06), annual temperature range (bio07), mean temperature of the wettest quarter (bio08), mean temperature of the driest quarter (bio09), mean temperature of the warmest quarter (bio10), mean temperature of the coldest quarter (bio11), annual precipitation (bio12), precipitation of the wettest month (bio13), precipitation of the driest month (bio14), precipitation seasonality (bio15), precipitation of the wettest quarter (bio16), precipitation of the driest quarter (bio17), precipitation of the warmest quarter (bio18), precipitation of the coldest quarter (bio19), and elevation (Elev). The bioclimatic variables had a spatial resolution of 5 arc-min. The 20 variables were screened before processing by the MaxEnt model to avoid oversaturated prediction results. The distribution point data were imported into a file of 20 variables and subjected to multicollinearity analysis, followed by Pearson correlation analysis with SPSS 26.0 software. The MaxEnt model needed the spatial distribution point data of the predicted object and various environmental variable data for use in the prediction process. Not all environmental variable data played a decisive role in the prediction results, so it was necessary to screen the environmental variables in the MaxEnt model so that those that contribute less to the prediction results were eliminated and those that contribute more to the prediction results were retained. The environmental variables with a correlation greater than 0.8 and a small contribution were discarded.

### 2.3. Software and Map Data

The MaxEnt software (version 3.3.3) used in this study was downloaded from the MaxEnt homepage (http://www.cs.princeton.edu/~schapire/maxent accessed on 10 April 2023). ArcGIS 10.2 developed by the Environmental Systems Research Institute of the Unites States was used. The vector map of China at a scale of 1:4 million was downloaded from the Resource and Environment Science and Data Center of the Chinese Academy of Sciences (https://www.resdc.cn/).

### 2.4. MaxEnt Model Optimization and Establishment

The potential distribution regions of tomato leafminer were predicted by MaxEnt 3.4.1 software, and the MaxEnt model was optimized by adjusting the model’s feature combination (FC) and regularization multiplier (RM) parameters. Using the kuenm package in R, we set the RM range to [0, 4.0], with an increment size of 0.1, and used 29 FCs, namely, L, Q, P, T, H, LQ, LP, LT, LH, QP, QT, QH, PT, PH, TH, LQP, LQT, LQH, LPT, LPH, QPT, QPH, QTH, PTH, LQPT, LQPH, LQTH, LPTH, and LQPTH. A total of 1160 parameter combinations were screened. Using the logarithm of AICc as the evaluation index, the combination of parameters for the prediction model that minimized the logarithm of AICc was selected as optimal (Figure 2).

The processed tomato leafminer sample points and climate variables were imported into the MaxEnt model. The variable contribution rate was calculated using the jack-knife method, with an output format of “Logistic”, an output file type of “asc”, an operation type of “Crossvalidate” (i.e., the cross-validation method), a randomly selected test set of “25”, a number of repeated training cycles of “10”, and other parameters set to default values. The random test percentage was 25%, which meant that 75% of the total database was used as the random sample to train the model, and the other 25% of the total database was used to test the model predictions. The regulation frequency was “3.5”; the feature combination as “Quadratic features”, “Product features”, and “Threshold features”.

### 2.5. Division of Suitable Habitats of Tomato Leafminer

The results obtained by 10 runs of the MaxEnt Model were imported into ArcGIS software and the model results were classified and graded. The natural break point classification method (Jenks) function was used to divide the suitable habitat index (SHI), and the suitable habitats for tomato leafminer were divided into four levels in terms of SHI: unsuitable habitat, SHI < 0.172; poorly suitable habitat, 0.172 ≤ SHI < 0.390; moderately suitable habitat, 0.390 ≤ SHI < 0.532; and highly suitable habitat, 0.532 ≤ SHI.

### 2.6. Accuracy of the Prediction Results of the MaxEnt Model

The evaluation of the prediction results of the MaxEnt model showed that the data sample omission rate basically agreed with the predicted omission rate, indicating the good quality of the constructed model. Testing the accuracy of the distribution results of tomato leafminer’s suitable habitats predicted by the MaxEnt model by using the receiver operating characteristic curve analysis, the test AUC value was 0.879 (Figure 3). The accuracy tests under the four climate models showed that the AUCs of the MaxEnt model on the training set were all greater than 0.8 (Table 1). The AUCs of the MaxEnt model all being close to 1 indicated that the prediction results were satisfactory. The AUC was an indicator of the predictive accuracy of a model. Its value ranged from 0 to 1. The larger the value, the better the predictive performance, indirectly reflecting the predictive ability of the model. AUC equal to 1 meant that the distribution regions predicted by the model fully coincided with the actual distribution regions of the species, which is an ideal situation. The ROC curve was evaluated using the following criterion: the prediction result was unacceptable (fail) when AUC had a value of 0.5–0.6, acceptable (poor) at 0.6–0.7, average (fair) at 0.7–0.8, satisfactory (good) at 0.8–0.9, and very satisfactory at 0.9–1.

## 3. Results

### 3.1. Determination of the Dominant Environmental Factors Affecting the Distribution of Tomato Leafminer

As calculated by the MaxEnt model, different environmental variables make different contributions to the potential distribution of tomato leafminer habitats (Table 2). The annual mean temperature was the main environmental factor limiting the distribution of tomato leafminer, contributing to 78.5% to the variability, while other influencing factors (mean diurnal range, precipitation of the coldest quarter, elevation, annual precipitation, precipitation of the driest month, and temperature annual range) had a combined contribution of 21.4%, all of these adding to a contribution of 99.9%.

The response curve associated between the environmental variables and the presence of tomato leafminer in this study is indicated. When the annual average temperature (bio01) was at 6.67~20.02 °C, the existence probability of tomato leafminer presence exceeded 0.5. When the average temperature daily difference (bio02) was at 7.20~14.87 °C, the probability of tomato leafminer existence exceeded 0.5. When the precipitation in the coldest quarter (bio19) was greater than 152 mm, the probability of tomato leafminer existence exceeded 0.5. At the elevation of (Elev) of 634.40~56.32 m, the existence probability of tomato leafminer exceeded 0.5. When the annual precipitation (bio12) was greater than 553.68 mm, the probability of tomato leafminer existence exceeded 0.5. When the precipitation in the driest month (bio14) was at 37.46~117.54 mm, the existence probability of tomato leafminer exceeded 0.5. With poor annual temperature (bio07) at 6.08~24.78 °C, the probability of tomato leafminer existence exceeded 0.5 (Figure 4).

Based on the regularization training gain in the MaxEnt model (Figure 5), the highest regularization training gain was the highest annual average temperature (Bio01), indicating that this variable provided more effective information for the prediction of tomato leafminer. In addition, the shortest was the annual average temperature (Bio01), which indicated that Bio01 had more unique information and was important for species distribution.

### 3.2. Distribution of Potential Suitable Habitats for Tomato Leafminer under the Current Climate

From the current potential suitable area of tomato leafminer (Figure 6), the suitable habitats for tomato leafminer are widely distributed in China: most of North China, most of East China, most of South China, most of Central China, most of Southwest China, some parts of Northeast China, and only a few parts of Northwest China (Figure 4). In particular, there are highly suitable habitats in Hebei, Shanxi, Jiangsu, Zhejiang, Fujian, Shandong, Taiwan, Guangdong, Guangxi, Henan, Hunan, Sichuan, Yunnan, Guizhou, Liaoning, Shaanxi, and Ningxia; in most of Beijing, Tianjin, and Chongqing; and sporadically in a few parts of Tibet, Inner Mongolia, Xinjiang Uygur Autonomous Region, Hubei, Jiangxi, Anhui, and Hainan. There are moderately suitable habitats in most of Anhui, Jiangxi, Hubei, and Hainan, and in a few parts of Inner Mongolia, Xinjiang Uygur Autonomous Region, Gansu, and Taiwan. The poorly suitable habitats are in some parts of Jilin, Inner Mongolia, Xinjiang Uygur Autonomous Region, and Gansu and, sporadically, in Tibet, Qinghai, and Heilongjiang.

### 3.3. Changes in the Suitable Habitats for Tomato Leafminer under Future Climates

All four climate models project a spread of tomato leafminer’s suitable habitats to the north and east of continental China under future climate conditions, with an increase in the size of suitable habitats in Inner Mongolia and the three northeastern provinces (Heilongjiang, Jilin, and Liaoning). Under the climate model SSP1–26, from 2021 to 2080, it is predicted that the highly suitable habitats for tomato leafminer will increase in the northern and northeastern provinces and will be sporadically distributed in the southeastern coastal areas. By 2100, the size of highly suitable habitats for tomato leafminer is projected to increase in the southern coastal areas of China, and the moderately suitable habitats in Guangxi, Guangdong, Fujian, and Zhejiang will mostly become highly suitable habitats. From SSP1–26 to 2100, the area will increase, with an increase of 1.87% (Table 3) (Figure 7).

The SSP2–45 climate model predicts an expansion of the suitable habitats for tomato leafminer (Figure 8), a gradual spread of highly suitable habitats to central Inner Mongolia and Northeast China, and a gradual shrinkage in highly suitable habitats in the southeastern coastal areas, shifting to moderately suitable habitats with a sporadic distribution of poorly suitable habitats. From SSP2–45 to 2100, the suitable area of tomato leafminer will increase compared with the current area, with an increase of 6.57% (Table 4).

Under the SSP3–70 climate model, the size of suitable habitats for tomato leafminer will expand (Figure 9), highly suitable habitats will constantly spread to northeastern China, and highly suitable habitats will gradually shrink in the southeastern coastal areas, where poorly suitable habitats will emerge during 2081–2100. From 2021 to 2100, the size of unsuitable habitats for tomato leafminer will decrease year by year. From SSP3–70 to 2100, the suitable area of tomato leafminer will increase compared with the current area, with an increase of 14.75% (Table 5).

Under the SSP5–85 climate model, it is predicted (Figure 10) that the size of highly suitable habitats for tomato leafminer will gradually decrease, that the highly suitable habitats will shift northeastwards from 2021 to 2080, and that the highly suitable habitats in the southeast coastal areas will shrink year by year. However, during 2081–2100, the highly suitable habitats will shift northwestwards, and poorly suitable habitats will emerge in the southeast coastal areas. The unsuitable habitats will gradually shrink and, by 2100, will make up a small portion of the northwest and northeast regions. From SSP5–85 to 2100, the suitable area of tomato leafminer will increase compared with the current area, with an increase of 24.25% (Table 6).

## 4. Discussion

The MaxEnt model was constructed from species distribution points and background environmental variables based on probability theory and machine learning theory, and the species distribution that it modeled was between potential and realistic distributions [30,31,32]. The results of the MaxEnt model were influenced by the complexity and sample size of the model. Models constructed with higher complexity have a better predictive performance, smoother response curve, and higher model transferability, so they can more reasonably reflect the responses of species to environmental factors and more accurately model the potential distributions of species [33]. The choice of environmental variables has an impact on the prediction results of ecological niche models [34,35]. The prediction of the geographical distribution of species using the MaxEnt model was implemented by inputting 19 bioclimatic factors from the Bioclim dataset of the World Climate Database (https://www.worldclim.org/) as the main or only environmental variables. The inevitable autocorrelation and multicollinearity between these 19 environmental variables were calculated according to specific needs [36]. The environmental variable factors with low contribution rates were eliminated, and those with high contribution rates were retained, thereby reducing the influence of redundant information on the prediction results of the geographical distribution of species and improving the accuracy of the prediction results. The sample size also affects the results of the MaxEnt model. The accuracy is unstable when the sample size is small, and as the sample size increases (towards a sample size of approximately 50 and 120 for the training data and the test data, respectively), the predictive accuracy of the MaxEnt model becomes increasingly stable [37]. This study had a sample size of 189, which, therefore, had no impact on the MaxEnt prediction results.

The present study provides evidence that host plant, temperature and rainfall are factors that greatly influence key biological parameters of *T. absoluta* [38]. Under natural conditions, insects and the growth and development of host plants are also mainly affected by rainfall and temperature. Therefore, the selection of environmental variables considers the temperature during the growth and development of tomato leafminer, the growth and development of plant, and rainfall. Insects in particular use rainfall cues to navigate the environment [39].

Temperature is an important factor affecting insect colonization, geographical distribution, occurrence number, life history, and behavior, making temperature a key abiotic factor influencing the success of biological invasion and the direction of insect spread. As an invasive pest in China, tomato leafminer has strong adaptability. The developmental threshold temperature, developmental maximum temperature, and developmental optimal temperature of tomato leafminer are 12.46, 30.40, and 27.36 °C, respectively [40]. As global warming continues, China will be a sensitive area for global climate change, with a higher warming rate than the global average over a given period. From 1951 to 2021, the annual mean surface temperature in China showed a significant upward trend. According to the Bulletin of the State of China’s Ecological Environment 2021 issued by the Ministry of Ecology and Environment, the annual mean temperature in China in 2021 was 10.53 °C, which is close to the developmental starting temperature of tomato leafminer, which will facilitate its spread [41]. The gradual increase in temperature and the use of warming equipment such as greenhouses in northern China during the winter provide overwintering sites for the pest, enabling it to spread northward. 

The tomato leafminer, *Tuta absoluta*, is a damaging pest of tomato crops worldwide. In the UK *T. absoluta* is controlled using an integrated pest management (IPM) strategy [42]. We can adopt the British approach, given its transmission and spread mode, occurrence characteristics, and harmful habits, so as to prevent the further spread of tomato leafminer and ensure the healthy development of the Solanaceae plant industry in China.

We used four shared economy models to predict the fitness area of tomato leafminer and five models to simulate the development path of future environment. With SSP1: sustainability—taking the green road, the world shifts gradually, but pervasively, toward a more sustainable path, emphasizing more inclusive development that respects perceived environmental boundaries. Land use is strongly regulated, e.g., tropical deforestation rates are strongly reduced. With SSP2: middle of the road, land use change is incompletely regulated, i.e., tropical deforestation continues, although at slowly declining rates over time. Rates of crop yield increase declines slowly over time. With SSP3: regional rivalry—a rocky road, land use change is hardly regulated. Rates of crop yield increase declines strongly over time, especially due to very limited transfer of new agricultural technologies to developing countries. With SSP5: fossil-fueled development—taking the highway, driven by the economic success of industrialized and emerging economies, this world places increasing faith in competitive markets, innovation, and participatory societies to produce rapid technological progress and development of human capital as the path to sustainable development. Land use change is incompletely regulated, i.e., tropical deforestation continues, although at slowly declining rates over time. Crop yields are rapidly increasing [43]. 

In the 2081–2100 SSP5–85 scenario, more than 90% of the suitable area of tomato leafminer grew too fast compared with other scenarios. It may be that in the context of SSP5–85, due to the rapid development of industry and the large emission of some greenhouse gases, global warming intensified, and the temperature increase in some areas with high altitude and low temperature in China gradually changed from non-suitable areas to suitable areas. In the context of 2021–2100SSP1–26, the percentage of the suitable area of tomato leaf moth only increased by 1.87%. This model is also an ideal model. Human beings have certain restraints on industrial development, leading to the slow occurrence of global warming and little temperature change, which led to little change in the suitable area of tomato leaf moth. However, this also reflects the phenomenon of tomato leafminer’s diffusion in the suitable areas of China, which also has certain reference significance.

## 5. Conclusions

This study used the MaxEnt model to predict the potential geographic distribution of tomato leafminer in China, using the publicly reported distribution of regions already invaded by tomato leafminer in China. The present study differed from previous predictions and forecasts of suitable habitats for tomato leafminer in China. Specifically, Luo et al., used ENM tools to eliminate highly correlated factors to reduce the influence of residual factors on the model; used the MaxEnt model to predict suitable habitats for tomato leafminer in China in between 2021 and 2080 under the SS1–26, SS2–45, and SS5–85 scenarios; predicted the centroid migration under the four coupled models SSP1–26, SSP2–45, SSP3–70, SSP5–85; and predicted changes in the suitable habitats for tomato leafminer in China from 2021 to 2100 under the four models [22]. However, the number of global distribution points after their screening was 66. Compared with the distribution points in this paper, there were no distribution points in many places. They may have missed distribution points, which would have had a certain impact on the results. They found that the current highly suitable habitats for tomato leafminer were mainly distributed in the eastern part of Sichuan, bordering Chongqing, Gansu, Shaanxi, Yunnan, and Guizhou, and that the pest will tend to spread toward higher latitudes in the future. In their results, there was no Xinjiang Uygur Autonomous Region where the tomato leafminer was first found in China. However, our study predicted that there are still some areas in the growing region in China, and the growing area in Xinjiang Uygur Autonomous Region has been increasing. 

In our study, after screening the relevant environmental variables by Pearson correlation analysis, the prediction model was optimized by adjusting its FC and RM parameters, and the model parameters leading to the best fit were selected. The MaxEnt model with optimally adjusted parameters can accurately reflect the response of a species to environmental factors. This study optimized the MaxEnt model to predict the distribution of the current suitable habitats of tomato leafminer and forecast the distribution of suitable habitats between 2021 and 2100 under the SS1–26, SS2–45, SS3–70, and SS5–85 scenarios. Our results were not consistent with the distribution of suitable habitats predicted by Luo et al., Currently, the tomato leafminer’s highly suitable habitats are mainly in most of North China, most of East China, most of South China, most of Central China, most of Southwest China, and a few parts of Northwest China. In the future, the suitable habitats for tomato leafminer will gradually spread to the high latitudes of northern China as well as to eastern and northwestern China. At present, tomato leafminer has scattered habitats in northwestern China. However, with the gradual increase in temperature in the future, the highly suitable habitats will shift from the southeast coastal areas to the northern high-latitude areas, the southeast coastal areas will change from highly suitable habitats to moderately suitable habitats, and more of northern China will become highly suitable, so the tomato leafminer will mainly spread to Northeast China, Inner Mongolia, and Xinjiang Uygur Autonomous Region. Based on the optimization of the MaxEnt model, this study mainly considered the influence of climatic factors on tomato leafminer but did not consider the influence of nonclimatic factors. Future research should focus on the influence of nonclimatic factors on the distribution of suitable tomato leafminer habitats.

## Figures and Tables

**Figure 1 insects-14-00531-f001:**
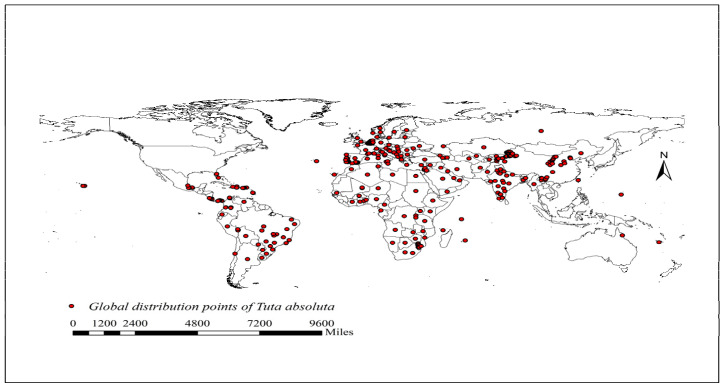
Distribution map of tomato leafminer across the world under current climate conditions.

**Figure 2 insects-14-00531-f002:**
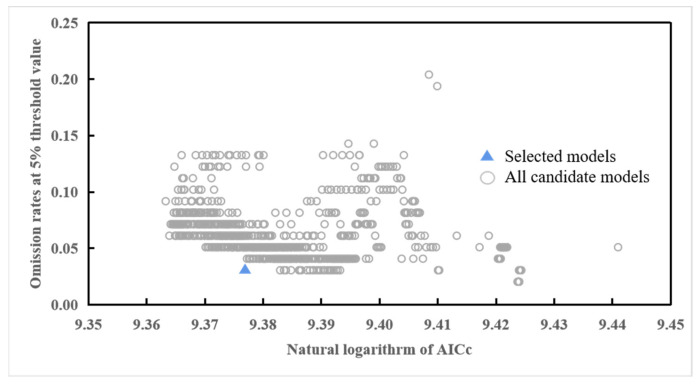
Model parameter selection diagram.

**Figure 3 insects-14-00531-f003:**
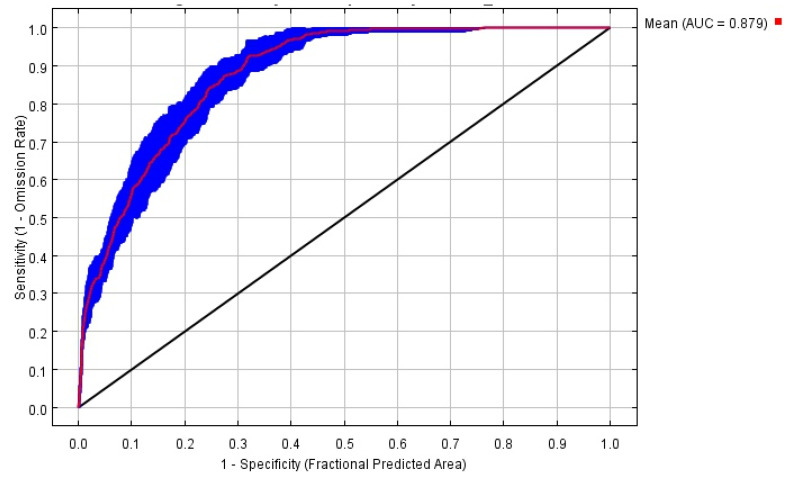
Reliability test of the distribution model created for tomato leafminer.

**Figure 4 insects-14-00531-f004:**
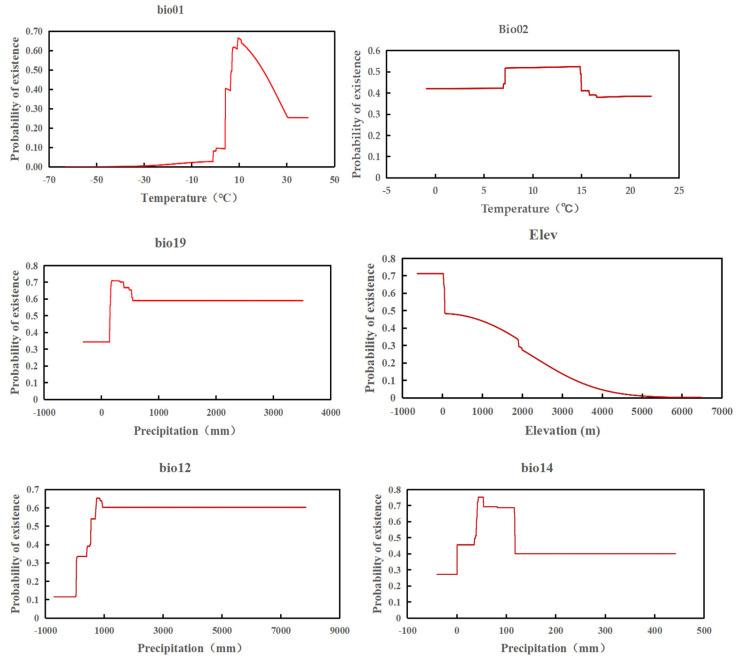
Response curves between probability of presence and climate variables.

**Figure 5 insects-14-00531-f005:**
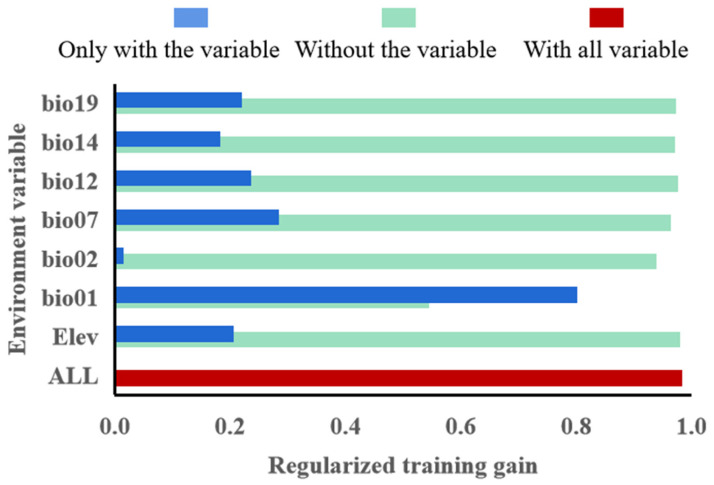
Importance of selected climate factors to MaxEnt model.

**Figure 6 insects-14-00531-f006:**
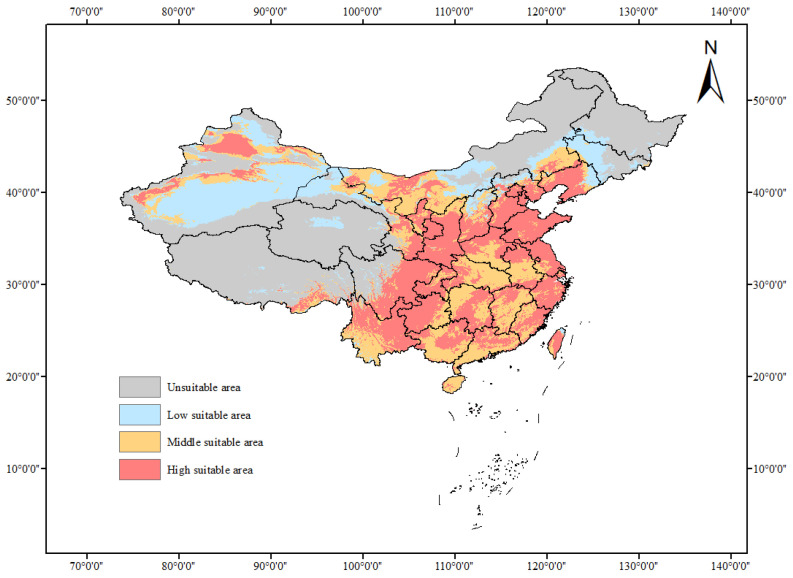
Current potential distribution of tomato leafminer in China.

**Figure 7 insects-14-00531-f007:**
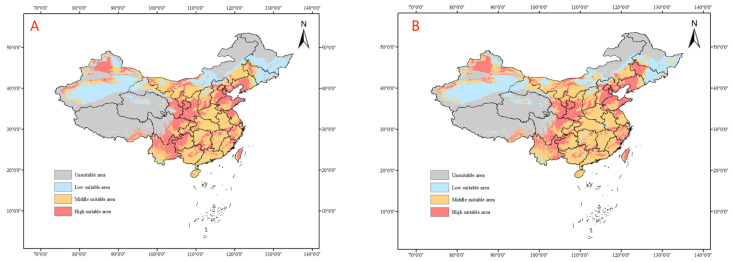
Predicted distribution of tomato leafminer in China from 2021 to 2100 under SSP1-26 ((**A**): 2021–2040, (**B**): 2041–2060, (**C**): 2061–2080, (**D**): 2081–2100)).

**Figure 8 insects-14-00531-f008:**
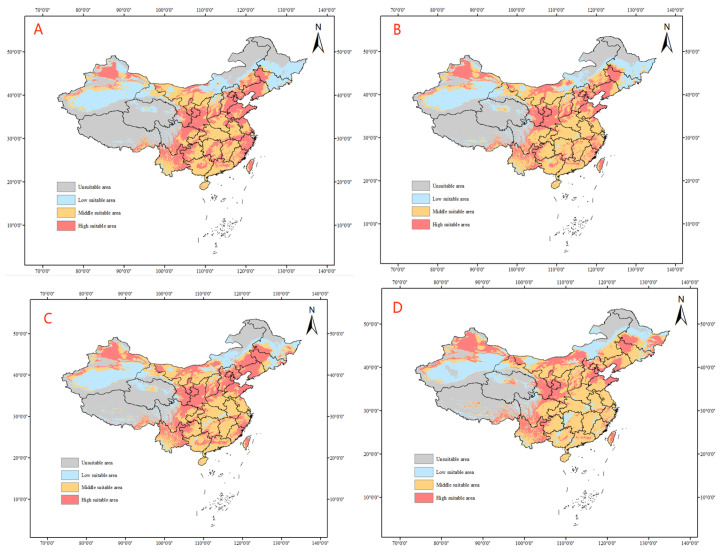
Predicted distribution of tomato leafminer in China from 2021 to 2100 under SSP2-45 ((**A**): 2021–2040, (**B**): 2041–2060, (**C**): 2061–2080, (**D**): 2081–2100)).

**Figure 9 insects-14-00531-f009:**
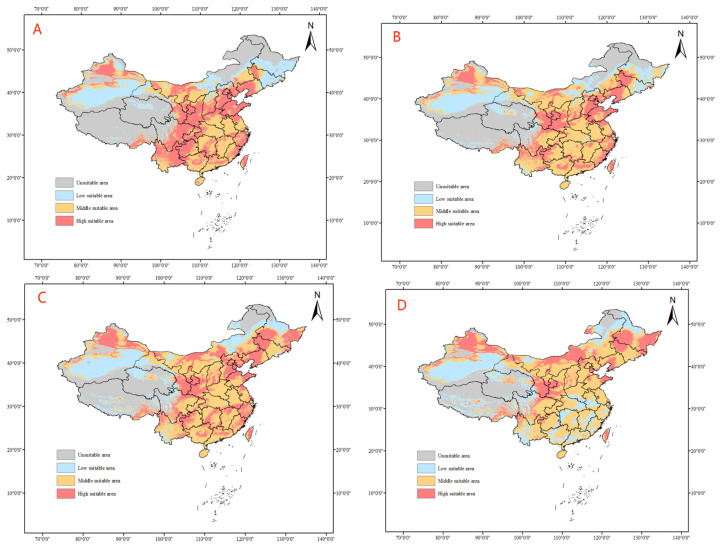
Predicted distribution of tomato leafminer in China from 2021 to 2100 under SSP3-70 ((**A**): 2021–2040, (**B**): 2041–2060, (**C**): 2061–2080, (**D**): 2081–2100)).

**Figure 10 insects-14-00531-f010:**
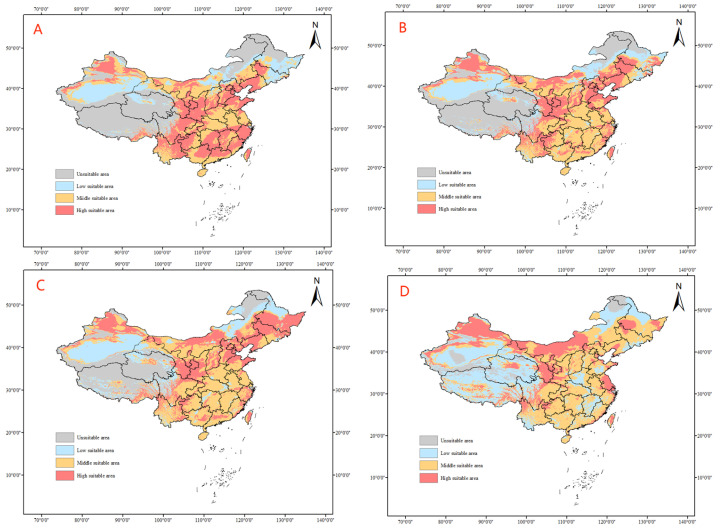
Predicted distribution of tomato leafminer in China from 2021 to 2100 under SSP5-85 ((**A**): 2021–2040, (**B**): 2041–2060, (**C**): 2061–2080, (**D**): 2081–2100)).

**Table 1 insects-14-00531-t001:** AUCs of tomato leafminer distribution prediction under different future climate models.

Climate Model for Prediction	AUC of the Training Set
2021–2040SSP1–26	0.831
2021–2040SSP2–45	0.835
2021–2040SSP3–70	0.831
2021–2040SSP5–85	0.827
2041–2060SSP1–26	0.830
2041–2060SSP2–45	0.834
2041–2060SSP3–70	0.829
2041–2060SSP5–85	0.836
2061–2080SSP1–26	0.835
2061–2080SSP2–45	0.829
2061–2080SSP3–70	0.826
2061–2080SSP5–85	0.830
2081–2100SSP1–26	0.829
2081–2100SSP2–45	0.831
2081–2100SSP3–70	0.834
2081–2100SSP5–85	0.834

**Table 2 insects-14-00531-t002:** Contribution rates and permutation importance (%) of environmental variables affecting the distribution of tomato leafminer.

Environmental Variable	Variable Name (Unit)	Percent Contribution (%)
bio01	Annual mean temperature(°C)	78.5
bio02	Mean diurnal range (°C)	6.5
bio19	Precipitation of the coldest quarter (mm)	4.8
Elev	Elevation (m)	3.9
bio12	Annual precipitation (mm)	2.8
bio14	Precipitation of the driest month (mm)	1.7
bio07	Temperature annual range (°C)	1.7

**Table 3 insects-14-00531-t003:** Area prediction of tomato leafminer in China from 2021 to 2100 under SSP1-26.

Different Prediction Periods	Low Suitable Area (km^2^)	Medium Suitable Area (km^2^)	High Suitable Area (km^2^)	The Total Area of the Suitable Area (km^2^)	Placement Area Proportion (%)
2021–2040SSP1–26	1,590,532.95	2,791,300.26	1,963,439.47	6,345,272.69	66.10
2041–2060SSP1–26	1,587,506.96	3,046,071.88	1,840,497.56	6,474,076.40	67.43
2061–2080SSP1–26	1,628,643.22	2,946,646.28	1,866,365.51	6,441,655.00	67.10
2081–2100SSP1–26	1,655,721.57	2,814,712.83	2,054,703.53	6,525,137.93	67.97

**Table 4 insects-14-00531-t004:** Area prediction of tomato leafminer in China from 2021 to 2100 under SSP2-45.

Different Prediction Periods	Low Suitable Area (km^2^)	Medium Suitable Area (km^2^)	High Suitable Area (km^2^)	The Total Area of the Suitable Area (km^2^)	Placement Area Proportion (%)
2021–2040SSP2–45	1,720,789.14	2,493,438.45	2,196,890.79	6,411,118.38	66.78
2041–2060SSP2–45	1,637,911.41	3,121,082.01	1,919,484.72	6,678,478.14	69.57
2061–2080SSP2–45	1,627,761.36	2,885,555.73	2,390,122.28	6,903,439.37	71.91
2081–2100SSP2–45	1,585,138.03	3,469,261.82	1,987,388.08	7,041,787.93	73.35

**Table 5 insects-14-00531-t005:** Area prediction of tomato leafminer in China from 2021 to 2100 under SSP3-70.

Different Prediction Periods	Low Suitable Area (km^2^)	Medium Suitable Area (km^2^)	High Suitable Area (km^2^)	The Total Area of the Suitable Area (km^2^)	Placement Area Proportion (%)
2021–2040SSP3–70	1,540,076.62	2,331,037.54	2,478,896.38	6,350,010.54	66.15
2041–2060SSP3–70	1,522,508.55	3,306,549.66	1,915,697.90	6,744,756.11	70.26
2061–2080SSP3–70	1,633,917.10	3,353,565.00	2,349,383.72	7,336,865.82	76.43
2081–2100SSP3–70	2,160,094.67	3,824,876.93	1,781,810.52	7,766,782.12	80.90

**Table 6 insects-14-00531-t006:** Area prediction of tomato leafminer in China from 2021 to 2100 under SSP5-85.

Different Prediction Periods	Low Suitable Area (km^2^)	Medium Suitable Area (km^2^)	High Suitable Area (km^2^)	The Total Area of the Suitable Area (km^2^)	Placement Area Proportion (%)
2021–2040SSP5–85	1,401,849.10	2,527,865.65	2,607,008.42	6,536,723.17	68.09
2041–2060SSP5–85	1,544,434.06	2,986,675.89	2,390,001.24	6,921,111.19	72.09
2061–2080SSP5–85	1,625,772.85	3,396,222.91	2,516,920.18	7,538,915.84	78.53
2081–2100SSP5–85	2,876,529.61	3,885,569.78	2,102,911.98	8,865,011.38	92.34

## Data Availability

Data are available upon request from the authors.

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
