# Peer review of "Prediction of the Current and Future Distribution of Tomato Leafminer in China Using the MaxEnt Model"

_insects, 2023, doi:10.3390/insects14060531_

Round 1
Reviewer 1 Report
Dear Authors,
The study deals with actual and important topic. Text is understandable.
The authors noted that this is not the first attempt to model the distribution of Tuta absoluta for China. The discussion did not sufficiently indicate the differences between the previous results and the results obtained in this study. The discussion does not convey a clear message about the contribution to the knowledge about the species given by the presented research.
The authors wrote "Our results agree with the distribution of highly suitable habitats predicted by Luo et al.", so is the developed model really more perfect?
In my opinion, an important drawback, absolutely requiring improvement, is the lack of a list of locations with their source (citations), as an appendix. Location data should be cited in the same way as other research results, especially when they form the basis of the developed model.
There are also insufficiently explained elements in the methodology, which were indicated in the pdf file.
The work contains numerous, minor editing errors consisting in missing characters, e.g. spaces. Other notes were added directly to the manuscript.
It is also worth further elaborating the discussion and conclusions. For a species of such great economic importance, little specific information can be found in these parts of the work. I encourage you to improve the article because success is possible.
Best regards

Author Response
Thank you very much for your valuable feedback. We believe that your feedback has been very helpful to us and is necessary. We have carefully revised your feedback accordingly. Comments on our manuscript.As you will see from the point-by-point response below, we had addressed all points raised by the reviewers in our revised manuscript.The following changes were made(reviewers comments in black,our response in red):
Response to Comments.
Point 1:Only one author is marked. His work was equal with whose?
Response 1: Yang and Jiang is the co-lead author of this article.
Point 2:No spaces after the dot.
Response 2: Modified.
Point 3:No spaces after the dot
Response 3:Modified.
Point 4:There isn't significant differences between "Simple summary" and "Abstract", it is just a little bit shorter. If it's not required, I suggest you remove it. If this part is required redraft is necessary. It does not add value to the work, it will not make it much easier for the layman to understand the research.
Response 4:Modified.
Point 5:Included in the title, so unnecessary in keywords.Lack of insect's Latin name in the title and key words.
Response 5:Remove the MaxEnt, and add the Tuta absoluta.
Point 6:Not correct citation. In cited paper there is no such information.
Response 6:Has been modified, the quoted article without the second half of the text, deleted the text of the second half of the sentence.
Point 7:One or two more examples of the great treat would be appreciated. In the case of such a well-known and studied species, more accurate citation could be given.
Response 7:An example was added as suggested.
Point 8:The dot should be after the brackets.
Response 8:Modified.
Point 9:No spaces after the dot.
Response 9:Modified.
Point 10:I suggest to convert this value to USD and put it in brackets or change the main currencies. It will be much more readable for readers around the world.
Response 10:The economic amount has been converted to US dollars at the exchange rate.
Point 11:The subchapter is called the “Collection and processing of distribution data...”, but there is no information whether the data was processed, and if so, how? What steps have the authors taken to reduce over-sampling to which the maxent model is susceptible? For example down sampling prevents overestimating species occurrence in regions with higher sampling effort and decreased spatial autocorrelation of data.
Response 11:The tomato leafminer distribution data used in this study were obtained from three sources: (1) The species distribution data from the valid monitored data record model of tomato leafminer were downloaded from the global biodiversity information facility (GBIF) database (http://www.gbif.org/). (2) Data were collected from the literature. The literature on the occurrence of tomato leafminer in China was collected by searching the CNKI database, and the literature reporting the occurrence of tomato leafminer in other countries was obtained by searching the English-language literature. (3) The data publicly reported and recorded by Chinese governments were also collected.
After manual deletion of preserved specimen Preserved specimen, material reference Material citation and fossil specimen Fossil specimen, the distribution point data was screened. To reduce oversampling in some areas and undersampling in others, reuse the ENM Tools software to screen distribution point data, with a screening accuracy of 5 km, and 393 effective distribution points data were obtained (Fig. 1). The standard China map base map is derived from the Standard Map Service (http://bzdt.ch.mnr.gov.cn/index.html) of the Ministry of Natural Resources, with a ratio of 1:4 million.
Point 12:Problem with hyphenation between lines.
Response 12:Modified.
Point 13:Unnecessary space before the dot.
Response 13:Modified.
Point 14:What was the process of defining these ranges?
Response 14:The natural break point classification method (Jenks) function was used to divide the habitat suitability index SHI (Suitable Habitat Index), and the suitable habitats for tomato leafminer were divided into four levels in terms of suitable habitat index (SHI).
Point 15:The abbreviation SHI typically expands to the Species Habitat Index. In this case is something else? What is the difference?How ranges were defined in this case?
Response 15:The range values are generated automatically by using the natural break point classification method (Jenks).
Point 16:This excerpt is more a description of the methodology than a result of presented studies. I suggest moving to the appropriate chapter.
Response 16:This part has been transferred to the method part
Point 17:There is no data on climatic conditions on the maps, so the reader cannot draw such conclusions based on them. Looking at the maps, which indicate only the countries where tomato leafminer was found, and not the exact place of occurrence, such conclusions cannot be drawn. If it were possible, modeling would be unnecessary. This statement is incorrect.
Response 17:This statement is incorrect.Easy to mislead the reader, has been deleted from the sentence.
Point 18:Better quality is necessary. 96 dpi is definitely too low resolution.
Response 18:Has changed the image quality to offer higher quality images.
Point 19:Looking at the map and its title we should understand everything.Information about what country it is, and borders of other countries are recommended. China is not suspended in a vacuum.
Response 19:To make it easier for readers to better understand which specific country the map refers to, China has been marked below the picture
Point 20:Inconsistent nomenclature. Once the abbreviation, and once the full name. Please unify.
Response 20:Modified.
Point 21:be understandable even without the text of the publication. Looking at them, we know which map represents which period. The legend is not legible. I suggest moving the description of colors to the caption of the figure.
Response 21:In order to better distinguish which period, A-D has been marked in the upper left corner of the picture to facilitate readers to understand the time.
Point 22:I am not sure if this is a discussion of the results obtained. This is a description of the methodology, and the work is not a methodical work. It does not bring innovative modeling methods, but only uses the existing ones.
Response 22:This paragraph only discusses the influence of choosing the environmental variables and the effect of the sample size on the prediction results of the MaxEnt model.
Point 23:Lack of space before brackets.
Response 23:Modified.
Point 24:T. absoluta and italic.
Response 24:Already italic.
Point 25:Lack of citation [23].
Response 25:Modified.
Point 26:Lack of space before bracket in this and many more places.
Response 26:Modified.
Reviewer 2 Report
Summary
The manuscript includes a species distribution model with current and future projections of the tomato leafminer, Tuta absoluta. The model is optimized through model selection of the FC and RM as well as removing colinear environmental variables. Four future climate models (SSP1-26, SSP2-45, SSP3-70, and SSP5-85) at four time points are predicted for China. Suitable areas, in general, are expected to shift with time and the severity of climate change. Environment variables that contributed most to the model include annual mean temperature, isothermality, and mean diurnal range.
General comments
The manuscript uses an interesting and devastating crop pest, the tomato leaf miner. The introduction nicely highlights the importance of understanding the potentially suitable areas for the species.
Methods: I am glad you optimized your model by selecting the optimal FC and RM. This is often neglected when creating species distribution maps with MaxEnt. Thank you for your diligence. The manuscript would be stronger if response curves for each environment variable were included and analyzed. MaxEnt automatically produces these, so the addition should be manageable, and it will provide much-needed specifics for the discussion. Lastly, there are concerns about the global data used; please see comments about Figure 3.
Results: The colors used in your maps are aesthetically pleasing and readable for individuals with different versions of colorblindness. Including the percent of land cover of each suitable habitat for each model and any effect sizes between models will greatly strengthen the results.
Conclusions: There is a lack of comparison between this model and the others mentioned in the introduction. Also, there is little reference for how the model promotes the monitoring, early warning, prevention, and control of the pest, as suggested by the abstract. Including more reference material and specific biology based on what was modeled would be more informative than generic statements that humidity and temperature influence insect growth.
Generally, the writing is easy to follow, but please review the manuscript for errors. For instance, a space is missing before “Here,” in line 16, and “Under” in line 19, and throughout the paper. Additional spaces are also prominent (example in lines 76). Scientific names also need to be italicized correctly.
Specific comments
Lines 85-86: The statement “the establishment of the MaxEnt model mainly considers environmental variables related to temperature and humidity” is not entirely true. The MaxEnt model allows users to upload any variable, including biological variables, such as vegetation cover. Many users, however, utilize temperature and humidity along with precipitation because they are incorporated into future projection climate change models and provided as WorldClim data. The limited use of environmental factors is thus user-derived, not because of what MaxEnt considers.
Lines 88-90: The items in the list are not all models. For instance, BIOCLIM is monthly temperature and rainfall values that can be used in models. This sentence misleads readers that some are variables that can be added to models, not models themselves.
Lines 134-135: Stating how the four climate change scenarios differ will help inform the reader of the importance of your different models. This can also be repeated in the results and supported in the discussion.
Lines 136-147: The introduction highlights the importance of temperature and humidity on insect growth and distribution. However, the variables used in the model presented in the manuscript are temperature and rainfall. There is a disconnect between the introduction and the methods used. Precipitation/rainfall is a very important component of insect survivorship and occurrence. It is confusing why it was neglected in the introduction but is included as half of the environmental variables.
Line 156: How did you determine which variable contributed less to the model? For instance, did you use a jackknife test, and/or did you assess the response curves of each variable?
Lines 167-175: Which combination (for instance, LQH 1.0) was selected by AIC as optimal? This is needed to reproduce your results.
Fig 1: The colors in the key do not match the colors in the figure. I appreciate the change from the green line in the figure to the grey in the key. The color scheme in the key is more inclusive to those with color blindness.
Fig 1 and 2: The axis labels are small and difficult to read.
Methods based on Fig 3: There are quite a few data points worldwide. Several of these are in countries with only one data point, for instance, much of Africa and South America.
(1) How did you determine that these points are indeed location data and not museum specimens, which are also included in GBIF?
(2) Additionally, these regions may not be sampled as thoroughly or reported in the published literature, which is at least the case with other insect species. Uneven and biased sampling across geographic space can result in biases in the environmental space and species niche that are projected (see Kadmon et al. 2004). Have you projected suitability using a more limited data set that is more evenly sampled (for instance, limited to just Asia or Europe and Asia)?
(3) By including the global data, the model also comprises global background points. As you maintained this to be the default of only 10,000 points, several of these background points are likely located in regions that are not only less sampled but environmentally dissimilar. This may misinform and weaken the model (see Veloz 2009) as MaxEnt uses presence-background data to create pseudo-absences that estimate potential suitability (hence the lack of need to include absence data). With this in mind, is the global data set appropriate for your projections?
References:
Kadmon, R., Farber O., & Danin A. (2004). Effect of roadside bias on the accuracy of predictive maps produced by bioclimatic models. Ecological Applications, 14(2), 401-413. https://doi.org/10.1890/02-5364
Veloz, S.D. (2009). Spatially autocorrelated sampling falsely inflates measures of accuracy for presence-only niche models. Journal of Biogeography, 36(12), 2290-2299. https://doi.org/10.1111/j.1365-2699.2009.02174.x
Sections 3.3-3.4: Including the percent of land cover of each suitable habitat (unsuitable, poor, moderate, highly) for each model (current and all four SSP future forecasts) will greatly strengthen the results. Additionally, including an effect size (for instance, a 5% increase in the size of suitable habitats between….) will help the reader understand how much is changed between the models. Without this information, the models are less meaningful, and the reader must rely on their own (potentially incorrect) interpretation of the maps.
Figures 5-8: It needs to be clarified what each map is. Either title descriptions for each map or a more descriptive caption with each map labeled as A-D is needed. Additionally, the keys are tiny and not readable, even when zoomed in.
Lines 305-306: The model did not include relative humidity or plant species. Rainfall and temperature are the only environmental variables reported with WorldClim BioClim, which was used. Relative humidity can also be uploaded as a variable to MaxEnt, but it is not inherent in the 19 used in the current manuscript. The connections made between rain and humidity and plants in the following sentences are unsubstantiated with references or specifics. Instead, a much more direct and supported message would include response curves of each of the environmental variables in the results and a discussion of the insect’s biology of what was modeled.
Lines 314-317: Please provide references.
Line 320: What is meant by theoretical generation number? Is this generation time or reproduction fecundity?
Lines 314-334: Consider breaking apart this paragraph and fleshing out some specifics. Connect this more directly with the variables included in your model, not just temperature generally. Without assessing response curves, the statements here are made with the assumption that the potential suitability made by MaxEnt follows the development temperatures.
Generally, the writing is easy to follow, but please review the manuscript for errors. For instance, a space is missing before “Here,” in line 16, and “Under” in line 19, and throughout the paper. Additional spaces are also prominent (example in lines 76). Scientific names also need to be italicized correctly.
Author Response
Thank you very much for your valuable feedback. We believe that your feedback has been very helpful to us and is necessary. We have carefully revised your feedback accordingly. Comments on our manuscript.As you will see from the point-by-point response below, we had addressed all points raised by the reviewers in our revised manuscript.The following changes were made(reviewers comments in black,our response in red):
Response to Comments.
Point 1:Methods: I am glad you optimized your model by selecting the optimal FC and RM. This is often neglected when creating species distribution maps with MaxEnt. Thank you for your diligence. The manuscript would be stronger if response curves for each environment variable were included and analyzed. MaxEnt automatically produces these, so the addition should be manageable, and it will provide much-needed specifics for the discussion. Lastly, there are concerns about the global data used; please see comments about Figure 3.
Response 1: We produced and analyzed the response curves for each environmental variable.The tomato leafminer distribution data used in this study were obtained from three sources: (1) The species distribution data from the valid monitored data record model of tomato leafminer were downloaded from the global biodiversity information facility (GBIF) database (http://www.gbif.org/). (2) Data were collected from the literature. The literature on the occurrence of tomato leafminer in China was collected by searching the CNKI database, and the literature reporting the occurrence of tomato leafminer in other countries was obtained by searching the English-language literature. (3) The data publicly reported and recorded by Chinese governments were also collected.
After manual deletion of preserved specimen Preserved specimen, material reference Material citation and fossil specimen Fossil specimen, the distribution point data was screened. To reduce oversampling in some areas and undersampling in others, reuse the ENM Tools software to screen distribution point data, with a screening accuracy of 5 km, and 393 effective distribution points data were obtained (Fig. 1). The standard China map base map is derived from the Standard Map Service (http://bzdt.ch.mnr.gov.cn/index.html) of the Ministry of Natural Resources, with a ratio of 1:4 million.
Point 2:Generally, the writing is easy to follow, but please review the manuscript for errors. For instance, a space is missing before “Here,” in line 16, and “Under” in line 19, and throughout the paper. Additional spaces are also prominent (example in lines 76). Scientific names also need to be italicized correctly.
Response 2: Modified.
Point 3:Lines 85-86: The statement “the establishment of the MaxEnt model mainly considers environmental variables related to temperature and humidity” is not entirely true. The MaxEnt model allows users to upload any variable, including biological variables, such as vegetation cover. Many users, however, utilize temperature and humidity along with precipitation because they are incorporated into future projection climate change models and provided as WorldClim data. The limited use of environmental factors is thus user-derived, not because of what MaxEnt considers.
Response 3:Modified.
Point 4:Lines 88-90: The items in the list are not all models. For instance, BIOCLIM is monthly temperature and rainfall values that can be used in models. This sentence misleads readers that some are variables that can be added to models, not models themselves.
Response 4:Modified.
Point 5:Lines 134-135: Stating how the four climate change scenarios differ will help inform the reader of the importance of your different models. This can also be repeated in the results and supported in the discussion.
Response 5:We use four shared economy models to predict the fitness area of tomato leafminer, five models to simulate the development path of future environment,SSP1: sustainability—taking the green road: The world shifts gradually, but pervasively,toward a moresustainable path, emphasizing more inclusive development thatrespects perceived environmental boundaries. Land use is stronglyregulated,e.g. tropical deforestation rates are strongly reduced, SSP2: middle of the road: Land use change is incompletely regulated, i.e. tropical deforesta-tion continues, although at slowly declining rates over time.Ratesof crop yield increase decline slowly over time, SSP3: regional rivalry—a rocky road:Land use change is hardly regulated.Rates of crop yieldincrease decline strongly over time, especially due to very limitedtransfer of new agricultural technologies to developing countries, SSP5: fossil-fueled development—taking the highway: driven by the economic success of industrialized and emergingeconomies,this world places increasing faith in competitivemarkets,innovation and participatory societies to produce rapidtechnological progress and development of human capital as thepath to sustainable development. Land use change is incompletelyregulated, i.e. tropical deforestation continues, although at slowlydeclining rates over time. Crop yields are rapidly increasing
Point 6:Lines 136-147: The introduction highlights the importance of temperature and humidity on insect growth and distribution. However, the variables used in the model presented in the manuscript are temperature and rainfall. There is a disconnect between the introduction and the methods used. Precipitation/rainfall is a very important component of insect survivorship and occurrence. It is confusing why it was neglected in the introduction but is included as half of the environmental variables.
Response 6:The revised manuscript highlights the effect of rainfall on insects, and indeed the environmental variables represent a large proportion of precipitation
Point 7:Line 156: How did you determine which variable contributed less to the model? For instance, did you use a jackknife test, and/or did you assess the response curves of each variable?
Response 7:Yes we use a jackknife test and assess the response curves of each variable.(Fig. 4, Fig. 5).
Point 8:Lines 167-175: Which combination (for instance, LQH 1.0) was selected by AIC as optimal? This is needed to reproduce your results.
Response 8:A total of 1160 parameter combinations were selected, namely, L, Q, P, T, H, LH, LT, LT, QH, QH, QT, PT, TH, LH, LQH, LQT, LH, LQH, LPT, LT, LPT, QPH, QPH, QTH, LQPT, LQPH, LQTH, LPTH and LQPTH. Take the logarithm of AIC value as the evaluation index, and finally select the combination of model parameters with the smallest AIC log value (Fig. 2), which is the optimal prediction model parameters.
Point 9:Fig 1: The colors in the key do not match the colors in the figure. I appreciate the change from the green line in the figure to the grey in the key. The color scheme in the key is more inclusive to those with color blindness.
Response 9:Modified.
Point 10:Fig 1 and 2: The axis labels are small and difficult to read.
Response 10:Rethe higher quality pictures
Point 11:How did you determine that these points are indeed location data and not museum specimens, which are also included in GBIF?
Response 11:The website shows the source of coordinates: preserved specimens, material reference material references, and fossil specimens. After manual deletion of preserved specimen Preserved specimen, material reference Material citation and fossil specimen Fossil specimen, the distribution point data was screened.
Point 12:Sections 3.3-3.4: Including the percent of land cover of each suitable habitat (unsuitable, poor, moderate, highly) for each model (current and all four SSP future forecasts) will greatly strengthen the results. Additionally, including an effect size (for instance, a 5% increase in the size of suitable habitats between….) will help the reader understand how much is changed between the models. Without this information, the models are less meaningful, and the reader must rely on their own (potentially incorrect) interpretation of the maps.
Response 12:The area of low, medium, and high suitability zones, as well as the overall suitability zone area, have been listed and displayed, calculating the proportion of the total area for easy observation by readers .
Point 13:Figures 5-8: It needs to be clarified what each map is. Either title descriptions for each map or a more descriptive caption with each map labeled as A-D is needed. Additionally, the keys are tiny and not readable, even when zoomed in.
Response 13:Regarding the image, mark A-D to distinguish between different stages, and also indicate the meaning represented by A-D below the image .
Point 14:Lines 305-306: The model did not include relative humidity or plant species. Rainfall and temperature are the only environmental variables reported with WorldClim BioClim, which was used. Relative humidity can also be uploaded as a variable to MaxEnt, but it is not inherent in the 19 used in the current manuscript. The connections made between rain and humidity and plants in the following sentences are unsubstantiated with references or specifics. Instead, a much more direct and supported message would include response curves of each of the environmental variables in the results and a discussion of the insect’s biology of what was modeled.
Response 14:Response curves for each variable's environmental variable have been uploaded, while the environmental response curve has been analyzed.
Point 15:Line 320: What is meant by theoretical generation number? Is this generation time or reproduction fecundity?
Response 15:The theoretical world algebra was calculated based on formulas in the cited references and has not been verified, therefore this sentence has been deleted.
Point 16:Line 320: Lines 314-334: Consider breaking apart this paragraph and fleshing out some specifics. Connect this more directly with the variables included in your model, not just temperature generally. Without assessing response curves, the statements here are made with the assumption that the potential suitability made by MaxEnt follows the development temperatures.
Response 16:Complded details while increasing the effect of precipitation on insects, linked to the main variable in the model, temperature and precipitation.
Round 2
Reviewer 1 Report
Dear Authours,
I am glad that my comments have found your appreciation and have been implemented, but the work still needs to be improved.
1. In response to the review Authors wrote:
Point 4:There isn't significant differences between "Simple summary" and "Abstract", it is just a little bit shorter. If it's not required, I suggest you remove it. If this part is required redraft is necessary. It does not add value to the work, it will not make it much easier for the layman to understand the research.
Response 4:Modified.
, but I can’t find these modifications.
2. The authors did not address the essential allegations related to the work methodology and, more specifically, the lack of geolocation data. Still, I can not find a table with data used during the model development process. It is obligatory. All used for modelling coordinates have to be included with citations.
Dear Authors, please create a table with columns: 1. latitude, 2. longitude, and 3. data source (GBIF or article citation). You can add it as an appendix.
3. Still, there is no correct citation of the GBIF database. Here you can find how to do it: https://www.gbif.org/citation-guidelines
Information in the text "...(GBIF) database (http://www.gbif.org/)..." is not enaugh. Please appreciate the work of the creators of the base and do it in accordance with the guidelines. Citing other papers that you have used is also highly recommended.
4. Modified and added text fragments are written carelessly. Spaces are missing or redundant in many places, and in other passages, incorrectly break words between lines. I have indicated a few such cases, but it is the authors' role to take care of such details and carefully check the text before depositing it.
5. What happened that SHI ranges have changed?
6. A few minor comments have been added to the manuscript.

Reviewer 2 Report
Lines 131-132: This added paragraph contains several phrases in duplicate. These include preserved specimen, material reference/citation, and fossil specimen. This is difficult to read and it is unclear why there are multiples.
Lines 133-135: This paragraph and these lines are not clear. From what I understand, the authors used ENM Tools software to screen distribution point data in an attempt to reduce oversampling in some areas and undersampling in others. However, this seems to mostly thin the occurrences so that only one data point remains within a 5km zone. This does help limit oversampling by thinning the data; however, this does not address under-sampling on the global scale and thus why global data was used.
Fig 2: This is an interesting figure, but not needed in the context of your work. It still does not explain which of your 1160 models was selected, in other words, which of your models is the blue triangle selected model? Additionally, the figure axis is labeled as “Natural logarithrm of AICc”. Logarithm is misspelled here. The text (Line 189) specifies that the logarithm of AIC was used. Why is one AIC and the other AICc?
Lines 237-254: This is a much-needed addition to the first paper. The biology of the insect is much clearer.
Fig 5. If this is the jackknife results, that should be added to the figure caption.
Line 267: the first sentence is incomplete.
Fig 7: A decimal point is needed after Fig
Tables: The additional tables concerning area prediction are very helpful.
Lines 310-317: The text format changed here.
Lines 404-418: This information is needed in section 2.2 as it explains what each of the models are. Here in the discussion, it is appropriate to compare the different shared economy models to one another, such as lines 420-423. This should be expanded upon.
Discussion: The discussion does not connect the specific environmental variables nor geographic distribution results. The authors merely replaced humidity with rainfall. In some occasions, this does not make sense (e.g., line 381). In others, the reference used no longer applies (e.g., reference #36 line 383). Overall, the discussion only repeats basic information from other papers and does not clarify or draw conclusions about the work done in this manuscript.
Manuscript: Please reread the text and provide spaces when needed (e.g., lines 377, 381, 383, 405, 406, 407, 408, and most lines thereafter).
Some editing for grammar and readability is needed, particularly for the added sections.
Round 3
Reviewer 1 Report
Ad. response 1. For a column "origin" a legend or full names are highly recommended.
The potential reader does not know what kind of source is "CN" (me also). In the dataset, there are only 36 positions called "literature". Using someone else's data without a specific source is not a good practice. It is necessary to provide the source (full citation) of information for each of them. I also do not see anywhere in the text a reference and information that the data is included in the appendix.
Ad. Response 3. "We added citations as well as public government reports to the manuscript."
"Public authority reports" are not a citation indicator. Please do it correctly, according to the guidelines under the previously sent link. Apparently, the public report got it wrong.
Ad. Response 6:Images have been modified.
But the description is still insufficient.
Few more questions and suggestions:
1. Why did the authors use the data about the species' occurrence only from China? What grounds are for rejecting data from other areas of the species' occurrence? I can't find the answer to this question at work.
I see that optimization for China was a priority, but using geolocation data from the entire range of the butterfly could give more information about the climatic requirements of the species, which definitely could improve the model. If the species is still spreading in China, and only data of occurrence from China were used, the model will not take into account all the potential of the species.
The question is important, also in the context of the small amount of data on the basis of which the model was developed.
2. As a source of the occurrence data, "Public reporting by the Chinese government" is listed ("The data publicly reported and recorded by Chinese governments were also collected").
It is not clear where these data were published. Could you cite them or show their direct source (webpage, bulletin, public written statements)?
Author Response
Thank you very much for your question. These are very important questions, and we deeply apologize for them. Please see the attachment.
